# Analytical Techniques Applied to the Study of Industrial Archaeology Heritage: The Case of *Plaiko Zubixe* Footbridge

**DOI:** 10.3390/molecules27113609

**Published:** 2022-06-04

**Authors:** Ilaria Costantini, Kepa Castro, Juan Manuel Madariaga, Gorka Arana

**Affiliations:** Department of Analytical Chemistry, Faculty of Science and Technology, University of the Basque Country UPV/EHU, P.O. Box 644, 48080 Bilbao, Spain; kepa.castro@ehu.eus (K.C.); juanmanuel.madariaga@ehu.eus (J.M.M.); gorka.arana@ehu.eus (G.A.)

**Keywords:** industrial heritage, µ-Raman spectroscopy, µ-EDXRF, pigments, conservation state

## Abstract

In this work, micro-Raman spectroscopy and micro-energy-dispersive X-ray fluorescence spectroscopy (µ-EDXRF) were applied on microsamples taken from the *Plaiko Zubixe* footbridge (1927) located in Ondarroa (Basque Country, Spain) in order to investigate the original paint coating and make an evaluation of the conservation state before its restoration. Elemental and molecular images were acquired for the study of the compounds distribution. Some modern pigments such as phthalocyanine blue and green pigments, minium, calcium carbonate, Prussian blue, and hematite were identified. Barium sulfate and titanium dioxide were recognized as opacifier agents. Thanks to the study of the stratigraphies, it has been possible to determine the original paint layer, which includes lead white, ultramarine blue, carbon black, and barium sulfate. In addition, colorimetric analyses made it possible to know the CIELab values of the original layer in order to reproduce the original colour during the planned restoration work. The massive presence of chlorine detected by µ-EDXRF and the corrosion products of the rust layer, in particular akaganeite and hematite, highlighted the atmospheric impact in the conservation of the bridge because they were due to the effect of both marine aerosol and to the presence of acidic components in the environment coming from anthropogenic activity. This work demonstrated the usefulness of a scientific approach for the study of industrial archaeology heritage with the aim to contribute to its conservation and restoration.

## 1. Introduction

The archaeological industrial heritage is a relatively new concept, born in the 1970s, when the need to preserve the proofs of industrialization process after they had been fallen into disuse or abandoned was declared [1]. Indeed, the industrial heritage concerns a particular type of heritage that includes objects, infrastructures, and works created during the industrial revolution, mainly for practical rather than decorative purposes that have had a strong impact on the territory and some of which have now been recently declared historical heritage.

On the one hand, the conservation of the industrial heritage concerns individual objects that can represent a symbol for the city and its inhabitants. On the other hand, the rehabilitation of the industrial heritage can involve entire urban areas, producing an evident increase in tourism and promoting a social, environmental, and economic development of the cities [2,3]. Therefore, regardless of the social impact, the industrial heritage conservation is unquestionably a topical issue.

Its restoration could be carried out with scientific standards, respecting the original appearance and the economic and technological environment of the time, or by creating a new work, a replica of the existing. In other situations, although the original appearance of a work was different, the authorities can decide to restore the appearance that the work had in the last few years. These latter cases are mainly linked to social or cultural reasons.

However, a scientific diagnostic approach, through the use of diagnostic techniques, capable of gaining knowledge of the technologies and materials used, has not been widely adopted in this field of research as has been the case for other types of works of art [4,5,6]. Currently, there are not many examples of scientific research in the literature concerning the study of industrial heritage. One of the most recent is the work by Tissot et al. [7] on the paint coatings of three energy generators from the early 20th-century power plant at Levada de Tomar (Portugal) that shows the importance of applying a scientific-diagnostic method even for the study of objects belonging to the industrial revolution [5].

The conservation of the industrial cultural heritage is strongly influenced by the environment in which the object is located. Both metal and steel, of which the industrial heritage is mainly composed, are considered among the most resistant materials, and for this reason they have been used for the construction of bridges and infrastructures. Despite this, if they are located near sources of humidity or environments with high relative humidity values, they can suffer from faster oxidation processes over the years. In particular, a marine atmosphere is one of the most corrosive environments for metallic structures due to the influence of marine aerosol. It is composed by organic and inorganic matter dissolved in water and includes primary (PMA) and secondary aerosol (SMA) particles. The primary aerosol is composed by suspended sea water drops rich in chloride- ions, generated by the interaction between wind and waves on the surface of the sea, which are deposited on the terrestrial surfaces according to a dry or wet deposition process [8]. The high content of airborne chlorides, mainly in the form of NaCl or KCl, react extensively with iron materials [9].

In addition, it is well known that the presence of SO_2_ as well as the action of other acid gases, such as NO*_x_* and CO_2_, can cause the increase of the corrosion rate in metals through wet or dry deposition mechanisms [10,11]. In wet deposition, the atmospheric acid gases react with the humidity and/or rainwater, giving rise to their acidic aerosols (H_2_CO_3_, H_2_SO_4_, and HNO_3_). The acidic nature of the moisture film deposited on the surface generates first the oxidation and then the dissolution of the metal, accelerating the corrosion mechanisms and the consequent formation of nitrate, sulfate, and carbonate salts. In dry deposition, the atmospheric gases can react directly with solid particles deposited on the surfaces [12]. Moreover, in an urban site close to the coast, the marine aerosol is rich in airborne particulate matter including metals such as Pb, Cd, Cr, Mn, Cu, Mo, Rh, Ni, As, Ti, V, and Hg coming from combustion processes, traffic, and industrial activity [13]. In addition, factors such as turbulence of the air, chemical affinity between pollutants, and the material and reactivity of the pollutants can accelerate the deposition phenomena [10]. Although in some cases the corrosion products have a protective function [14,15,16], in coastal atmospheres the presence of certain corrosion products, such as akaganeite (FeO_0.833_(OH)_1.167_Cl_0.167_) [17], can accelerate the corrosion rate in metal works of historical interest.

Thus, the rehabilitation of the iron-building heritage has been necessary because of the deterioration produced by natural and anthropogenic factors that endangered their survival and their usefulness to the society for which they were designed. This is the case with the Ondarroa footbridge. The footbridge belongs to the tradition of mobile iron bridges built in numerous navigable channels. Its particularity of being one of the few remaining rotating bridges preserved today, the only one in Spain, makes it a unique architectural element worthy of being preserved and a symbol of the country. Due to its precarious state of conservation, a restoration intervention was planned, which also had the aim of restoring its original colour. 

Thus, the present work aims at identifying the original colour of the footbridge of Ondarroa with a view to its future restoration so that it would be possible to recover its original appearance, since the metallic structure has been subjected, as a whole, to various chromatic changes from its construction to the last interventions. In addition, the impact of marine aerosol and the harbour environment in the bridge will be documented by the characterization of different corrosive compound and biomarkers. 

For this purpose, a scientific diagnostic study was necessary. The study was carried out on six micro samples, five of them as cross-sections, by means of elemental (micro-energy-dispersive X-ray fluorescence spectroscopy) and molecular analysis (Raman spectroscopy) after a careful observation under an optical microscope. Colorimetric analyses were also performed to know the colour values of the different pigments used.

## 2. Results and Discussion

### 2.1. Characterization of the Paint Layers 

Two samples collected in different areas in the largest piece, which belongs to the low part of the railing (Appendix A, Appendix A) received in the laboratory, presented the same stratigraphic composition; therefore, only the results of one sample are shown below. Specifically, three different homogeneous layers, two outermost green ones (Appendix A) and one inner red/orange (Appendix A), were recognized by observing the samples with a stereomicroscope. 

Thanks to Raman analyses, it was possible to identify the compounds that characterize the different paint layers. The outermost layer (layer a in Appendix A) of dark green colour was composed of phthalocyanine green (C_32_H_3_Cl_13_CuN_8_, Raman bands at 688, 740, 744, 815, 977, 1079, and 1208 cm^−1^, Figure 1a) [18], while the intermediate layer (layer b in Appendix A) of lighter green colour was made mainly with the phthalocyanine blue pigment (C_32_H_16_N_8_Cu, Raman bands at 236, 257, 483, 594, 680, 747, 779, 832, 953, 1007, 1108, 1143, and 1193 cm^−1^, Figure 1b) [18]. In addition, in many Raman spectra a very weak peak at 1038 cm^−1^ was visible (Raman band marked in the red circle in Figure 1b). This band could belong to the yellow azo pigment PY100 used in small amounts, mixed with phthalocyanine blue, to obtain a green colour. Contrary to the green coloured layers, the innermost orange layer (layer c in Appendix A) was made by mixing two compounds, minium (Pb_3_O_4_ Raman bands: 122, 152, 224, 313, 390, and 548 cm^−1^) [19] and a smaller amount of barium sulfate (BaSO_4_, Raman bands: 453, 460, 618, and 987 cm^−1^, Figure 1c) [20] since all the Raman spectra recorded in this area showed the main features of both compounds. 

As can be seen in the Raman spectroscopy images (Figure 2) carried out considering the band with the highest intensity of each compound, the green layers consist of phthalocyanine green (Figure 2b) and phthalocyanine blue (Figure 2c), respectively. On the other hand, the presence of barium sulfate used in mixture with minium was confirmed due to the presence of the two compounds in the same pictorial layer, as is evident from the overlapping of the Raman images in Figure 2e,f. The presence of the yellow azo pigment was identified on two pictorial layers in mixture with phthalocyanine blue and even with minium and barium sulfate like in the Raman image shown in Figure 2d. The Raman map was obtained considering the band at 1038 cm^−1^, assigned to SO_3_^−^ symmetric stretch [21], in order to avoid the overlapping with barium sulfate since both compounds have a Raman bands in the same position (989 cm^−1^). 

Thus, in this part of the bridge, after applying red lead, barium sulfate, plus a yellow azo pigment, a green paint (phthalocyanine blue and yellow azo pigment) was applied, and over it a dark-green one (phthalocyanine green),which was the colour that is visible currently. The use of red lead, currently banned due to its toxicity, was probably employed as antioxidant paint, following the rules in the second half of 20th century [22].

Unlike the previous case, the subsamples that were collected from the piece that permit the movement of the bridge (Appendix A) showed a different stratigraphic composition from a first observation with the stereoscopic microscope. The first analysed subsample (SUBS-1b), collected from a green area, shows four homogeneous and well-defined layers (Appendix A). The outermost pictorial layer was entirely composed of phthalocyanine green (Appendix A). In the second one, white in colour, which probably represents the primer layer, rutile (α-TiO_2_, Raman bands a: 442 and 608 cm^−1^) [23] and calcium carbonate (CaCO_3_, Raman bands at: 282 and 1086 cm^−1^) [23] were detected, as seen in Appendix A. Additionally, the innermost green layer of a lighter shade was composed of a mixture of Prussian blue (Fe_4_ [Fe (CN)_6_]_3_, Raman bands a: 277, 364, and 530 cm^−1^) [24] and barium sulfate (Appendix A).

Another pictorial layer was recognized, previously applied, thanks to the Raman spectroscopy images showed in Figure 3. At first glance, this layer had the same hue as the previous one and only by observation with a stereoscopic microscope it was not possible to recognize the two different layers. However, the Raman images made it possible to distinguish one more pictorial layer, entirely composed of phthalocyanine blue (Figure 3g). Thus, in this sample four layers were found, respectively, from the oldest to the most recent: phthalocyanine green, rutile mixed with calcium carbonate, Prussian blue with barium sulfate, and phthalocyanine blue.

Consequently, the stratigraphy of this subsample showed that on a black layer (its stratigraphy is described in the following subsample), a phthalocyanine blue primer was applied (layer g in Figure 3), on which another layer was composed of Prussian blue and barium sulfate (layer e + f in Figure 3). Next, another white primer composed of rutile and calcium carbonate was applied (layer c + d in Figure 3), and finally, a green layer of phthalocyanine green (layer b in Figure 3), which was the one visible nowadays.

Raman analysis on the surfaces of the subsample (SUBS-2b) taken from a black area showed the presence of carbon black, homogeneously distributed throughout the entire surface. Underneath, there was a heterogeneous layer consisting mainly of hematite found in grains of different sizes (Fe_2_O_3_, Raman bands a: 222, 240, 290, 405, 490, and 608 cm^−1^, Appendix A) [25]. In addition to iron oxide, the analyses on this layer have made it possible to recognize at various points of the Raman spectrum characteristic of a material composed of silicon and carbon (similar to silicon carbide wire, SiC, with Raman bands at: 150, 763, 786, 795, and 964 cm^−1^, Appendix A) [26]. Raman bands belonging to barium sulfate were also identified in the same layer. The distribution of the three compounds is indicated in the Raman images in Appendix A.

The observation of one of the subsamples (SUBS-3b) taken in the piece shown in Figure 1b, which is part of the system that allowed the movement of the bridge, allowed us to recognize a more complex stratigraphy. This sample was collected in the lower area of the dark piece where probably the original painting could remain. The first outer layer was dark in colour applied over a heterogeneous red and white layer that presented grains of different sizes as in the piece previously described. Additionally, a thicker layer of black colour and another of green colour could be clearly recognized. According to the observation with the stereomicroscope, the first layer applied in the sample was a heterogeneous layer of grey colour with dark and blue grains as shown in Figure 4. The detail of the blue-greyish layer, whose thickness was around 1.5 mm, is showed in Figure 4c. In the lower part of the mentioned layer of blue-grey paint, remains of iron oxide flakes (reddish colour) detached from the metallic surface were also observed.

At first, the subsample was analysed by micro-energy dispersive X-ray fluorescence (μ-EDXRF) to study its elemental composition. The elemental maps of each element on the analysed sample are shown in Figure 5. The µ-XRF analysis indicated the presence of chlorine, iron, silicon, barium, and zinc, uniformly distributed in the outer part of the subsample. Surprisingly, chlorine was not detected in the green layer indicating the absence of phtalocyanine green in this case as the green pigment. Iron and silicon belong to the first interior layer where hematite and the silicon carbide compound were detected. On the other hand, some of these elements such as iron, chlorine, silicon, barium, and zinc did not belong to the exterior green paint layer, but rather they were elements trapped by the atmospheric particles of the marine aerosol that arrived at the bridge continuously. Although the marine aerosol is mainly composed of chlorides, it can also carry fine particles in suspension composed of beach sand (in this site, there is a beach close to the bridge) that mainly contributes to increase Si and Fe concentration on the surface. The evidence of iron in the lower area of the sample, on the other hand, refers to remains of the metal support that belongs to the bridge structure and this contributes to the mass fraction of Fe in the sample. The presence of barium inside the subsample was evident occupying much of its surface and it was the element with the highest concentration in the sample. Among the main elements of this interior, sulfur, lead, calcium, and zinc stand out. The semi-quantitative values of each element in the sample SUBS-3b are shown in Table 1.

In addition, in urban and harbour areas, other ions are also present in a suspended way such as Ba^2+^, Zn^2+^, Ca^2+^, K^+^, Mg^2+^, Fe^3+^, Al^3+^, Sr^2+^, NH_4_^+^, HCO^3−^, and Br^−^. The source of these anions and cations can reside in the influence of maritime traffic, port activities, and also industry or road traffic [27].

The molecular composition of the most layers was the same as in the previous subsample (SUBS-2b), since carbon black was identified on the outside and, underneath, there was a heterogeneous layer with hematite, barium sulfate, and the silicon carbide. This pigment composition accounts for the presence of the elements Si, S, Ba, and Fe identified by µ-EDXRF analysis. In this sample, the green colour was not related with the use of phthalocyanine pigments. Indeed, Prussian blue, the yellow pigment lead chromate (Raman bands at 337, 360, 376, 402, and 840 cm^−1^ Appendix A) and barium sulfate were the major components of the green colouration identified in the interior of the subsample. In addition, lead white (2PbCO_3_·Pb(OH)_2_) was found in some points of analysis in this area. Thus, the presence of lead was very irregular, and it represents the second main element, of the total sample after Ba. However, a major content of lead appears in the lower part of the sample (the one in contact with metallic iron) that could be associated with the use of lead white, identified mostly in the blue and grey layers (as will be seen later as well) and only in traces in the green layer. The low values of chromium in the XRF semi-quantification, belonging to lead chromate, was justified with the high absorption coefficient both of lead of the same pigment and of lead white used in the mixture. Additionally, by Raman spectroscopy other minor compounds were detected in that blue-green layer, such as gypsum (Raman bands at 412, 492, 617, 668 1008, and 1134 cm^−1^, Appendix A) [28] and anatase (β-TiO_2_, Raman bands at 140, 192, 393, 512, and 635 cm^−1^, Appendix A) [29].

In the oldest grey paint layer, in most of the analysed points, Raman spectra of barium sulfate were recorded, with all the characteristic bands (Raman bands: 453, 460, 618, 648, 987, and 1140 cm^−1^, Figure 6a) both in the matrix and in loose grains of different sizes. Additionally, in the black and blue grains, Raman spectra of carbon black (Raman bands: 1347, and 1602 cm^−1^, Figure 6b) and ultramarine blue (Al_6_Na_8_O_24_S_3_Si_6_, Raman bands: 260, 546, 583, 805, 1095, and 1644 cm^−1^, Figure 6c) [30] were recorded, respectively. In addition, lead white (Raman band at 1050 cm^−1^) was identified in the original paint layer, although in lesser quantity compared to the other compounds (Figure 6d). All the pigments found in the original paint layer of the footbridge are shown in the Raman spectra collected in Figure 6.

No chlorine compounds were found by Raman spectroscopy in this sample. The presence of this element, identified by µ-EDXRF only in the outermost part of the sample, is possibly due to the characteristics of the natural marine environment in which the bridge is located, as discussed later in detail.

The results allow reconstructing the execution of painting on the railing structure. Its original colour was greyish blue (see blue layer of Figure 4) composed of the mixture of ultramarine blue and carbon black. These pigments were mixed with lead white and anatase, with these latter ones probably being used as opacifiers [31]. On top of this greyish blue layer, another paint of a more greenish blue hue was applied (see greenish layer above the blue in Figure 4) composed of Prussian blue mixed with lead chromate and dispersed in barium sulfate (with traces of gypsum) and anatase. Over time, a hematite antioxidant primer was applied and over it, a black to cover possibly all rust formation and colour flakes.

### 2.2. Colorimetric Studies

Colour measurements in CIELab colour space were collected from the pieces of the bridge (Appendix A) delivered to the laboratory (Table 2) that permit the replication of colour. Therefore, the values of the green colour and of the original greyish blue colour were recorded, obtained by carefully scraping the surface of the piece that allows the rotation of the bridge.

### 2.3. Evaluation of the State of Conservation for the Iron Structure of the Bridge 

The Raman analyses were also applied for the study of corrosion products in the iron structure generated by the exposure of the bridge to the marine environment that favoured its disintegration. The state of conservation of the pieces delivered to the laboratory was different from each other; therefore, an oxidized chip (SUBS-4b) without a paint layer from the piece in Appendix A and a sample (SUBS-2a) with paint layer from the base of the railings (Appendix A) were selected. 

The Raman maps (Figure 7) carried out in the oxidized sample SUBS-2a, treated as cross-section, highlighted the presence of goethite (α-FeOOH, Raman bands: 250, 330, 390, 478, and 550 cm^−1^ as shown in Figure 8a) [32] as the main compound in the internal area of the oxidized chip, as seen in the Raman image shown in Figure 7a. The presence of magnetite (Fe_3_O_4_, Raman bands: 550 and 663 cm^−1^ shown in Figure 8c) [33] was also identified by point-by-point Raman spectroscopy, and its presence is important in the rust area as shown in the Raman image of Figure 7c. On the other hand, the presence of lepidocrocite (γ-FeO(OH), (Raman bands: 215, 249, 305, 345, 376, 523, 645, and 1300 cm^−1^ as depicted in Figure 8b) in the edges of the sample and in correspondence to microfractures of the subsample was identified as seen from the Raman image shown in Figure 7b. This observation is in good agreement with other investigations that documented the formation of lepidocrocite associated to local more aerated conditions [34,35]. This species of iron oxyhydroxide is known to be one of the most unstable forms of the corrosion compounds, which can transform into the more stable goethite with the succession of wet–dry cycles during the passivation of the corrosion processes [36]. This means that in the sample, the decay process has not been yet completed and is still going on. 

In agreement with our results, previous studies on rust surfaces on mild steel demonstrated that in marine environments, lepidocrocite develops preferentially on the outermost surface, irrespective of the chloride ion deposition rate, while magnetite and akaganeite (not found in this sample), an oxyhydroxide formed in chlorine rich atmospheres, mainly form near the base steel. In addition, according to Diaz et al. [37] with the increase of the exposition time, rust layers become thicker, and the lepidocrocite is partially transformed into goethite generating a stratified bilayer structure of rust consisting of a porous outer layer of lepidocrocite and an inner layer of compact goethite. 

As mentioned, even magnetite was identified in the subsample as a corrosion product. Its formation is commonly detected as a decay compound in rust developed in marine atmospheres, and it is usually detected in the inner zone closest to the base steel, where the lower oxygen availability favours its development [38,39].

Raman results revealed that the structure was also affected by biological colonization. The carotenoid pigment astaxanthin, recognized by the three main bands at 1509  cm^−1^ (ν_1_ C═C), 1152  cm^−1^ (ν_2_ C-C and 1001  cm^−1^ (ν_3_ C-H) and even by the overtones at 957, 1191, 1448, 2150, 2296, and 2650 cm^−1^, was detected in extended areas of the sample [40]. The spectrum in Figure 8b shows astaxanthin and lepidochrocite in the same spot area, showing how the colonization process extends also to the oxidized rusts. Among the carotenoid pigments, astaxanthin is the most oxidized species and is known to be synthesized by photosynthetic organisms such as cyanobacteria, fungi, and algae as a defense mechanism against atmospheric pollution. For this reason, it was proposed as bioindicators of high concentration of SO_2_ in the atmosphere [41]. In our study, the presence of astaxanthin in corrosion patina was probably related to the acidic environment in which the bridge was located. The industrial activity present in the outskirts of the city and the close proximity to the port are the responsible for the high concentration of this compound in the oxidized microsample.

Furthermore, surface analyses were carried out on a microsample (SUBS-4b) collected from an oxidized area of the sample that allowed the movement of the bridge (Appendix A). Only the presence of lepidocrocite, with a globular morphology, was identified in the inner surface of the metal fragment. On the other hand, the exterior side showed a more heterogeneous composition. The presence of akaganeite (FeO_0.833_(OH)_1.167_Cl_0.167_), a highly unstable Cl^−^ bearing corrosion phase, was detected in some black areas of the sample. This compound was characterized by its Raman bands at 310, 390, 535, and 724 cm^−1^ (Figure 8d), and its identification in the sample suggested the high impact of the chloride rich marine aerosol in the rotating bridge structure. As the akaganeite structure is characterized by tunnels partially occupied by chloride anions parallel to the c-axis of the tetragonal lattice, it tends to form low density, compared with other corrosion products, and fragile rust layers promoting cracking and exfoliation phenomena [42]. According to the investigation of Li et al. [43], once the akaganeite is saturated with Cl^−^ and it cannot take up any more Cl^−^, free Cl^−^ can be available to accelerate corrosion at anodic sites.

In the same spectrum showing akaganeite, the presence of black carbon (Raman bands: 1360 and 1600 cm^−1^ as seen in Figure 8d) was evident. This suggests an important amount of carbon in the small spot of the Raman observation (approximately 10 µm in diameter) that can only be explained by deposition of atmospheric particulate matter containing soot.

The presence of hematite (Figure 8e) was also identified in some points on the rust layer. This formation of hematite was not largely documented in the literature between the corrosion products. In the research of De la Fuente et al., it was identified from the ferrihydrite formation only in a rust layer exposed to an industrial and marine atmosphere. The presence of hematite could be related to the influence of acid rainwater, and an SO_2_-rich atmosphere could have favoured the transformation.

Thus, the identification of compounds such as hematite, carbon or the most oxidized form of carotenoid pigments reflect the environmental conditions that affected the iron of the bridge during the years. In fact, it is located in an area where SO_2_ emissions are generated by the burning of fuel oils from boats, or by gases produced by industries located nearby and transported by winds.

## 3. Materials and Methods

### Microsampling and Laboratory Instrumental Set Up

The Ondarroa footbridge, known as *Plaiako Zubixe* (the Bridge of the Beach in the local Basque language) was inaugurated in 1927 with the name *Pasarela de Alfonso XIII* and until the 1980s, due to its peculiarity as a revolving bridge, it allowed boats to pass through the Artibai river. Since 2016, the walkway has been closed to pedestrians for safety reasons, due to its serious state of deterioration due to corrosion caused by the marine environment in which it is located. In 2008, it was declared a Cultural Asset by the Basque Government since, due to its rotating system, it represents an exceptional example that remains today of the solutions adopted, in previous decades, to allow the transits of boats and vehicles in the estuaries.

Two metal pieces were received in the laboratory and collected from different parts of the bridge. The samples have been taken in some points (Appendix A) where different paint layers appeared using a scalpel. Two subsamples were taken for their stratigraphic study from the piece that belongs to the lower part of the bridge railing, which is green in colour on the outside and has a circular geometric decoration (Appendix A). Four subsamples were taken from the second piece that belongs to the base of the railings and that permitted the movement of the bridge (Appendix A). In this case, the piece was partially covered with a layer of green paint while other areas show a green/black tone, with extensive areas where no pictorial layer appears and only the highly degraded metallic material could be seen. All samples have been treated as cross-sections, but since two samples of the railing piece were identical from a compositional point of view, only the results of one sample were showed. All the samples collected on each piece and the techniques employed are summarized in Appendix A.

Six samples taken from the two pieces (Appendix A) were prepared as cross-sections to acquire complete information on their stratigraphy. For the cold encapsulation of the collected samples, an acrylic polymer based on methyl methacrylate was used. After encapsulating the samples, they were polished to obtain a completely smooth surface. The polishing of the cross-sections was carried out with the Metkon Forcipol 1 polisher (Barcelona Quálites, s.l.) using WS FLEX 18C and 16 waterproof sandpaper (P320-2000) and cloths for the last finish with diamond paste with a granulometry of 1 μm.

First, high resolution images of the fragments were obtained using an SMZ-U stereomicroscope (Nikon, Japan) coupled with a Nikon DigitalSight DS-L1 camera.

Then, Raman analyses were performed using an InVia micro-Raman confocal spectrometer (Renishaw, UK) coupled to a Leica DMLM microscope equipped with 5×, 20×, 50×, 50× (long distance), and 100× objectives and using 532 and 785 nm as the excitation laser source. The lasers were set at low power (no more than 1 mW) to avoid thermal transformation of the samples. Data acquisition was carried out using Wire 4.2 software (Renishaw). Spectra were acquired between 150 and 3200 cm^−1^ (mean spectral resolution 1 cm^−1^) and several scans were recorded for each spectrum in order to improve the signal-to-noise ratio (3-20 s, accumulations of 5-200). Around 40 Raman spectra were recorded for each sample. For samples with a more complex stratigraphy, nearly 50 spectra were collected.

The elemental distribution maps were acquired using an M4 TORNADO EDXRF spectrometer (Bruker Nano GmbH, Germany). The lateral resolution used for spectral acquisitions was 20 microns. For most of the elements, the selected line was the K_α1_, except for Ba for which its corresponding L_α1_ line was employed. In addition, to improve the detection of the lightest elements (Z > 11), the measurements were acquired under a vacuum (20 mbar) by means of a diaphragm pump MV 10 N VARIO-B. Data collection and interpretation were carried out by means of the software M4 TORNADO (Bruker Nano GmbH), whereas the quantitative analyses were carried out using the deconvolution M-Quant software package based on the application of fundamental parameters quantitative methods.

Colour measurements were made with a PCE-CSM 5 Trisstimulus colorimeter with a measurement spot size of 4 mm, measurement geometry in the integrating sphere with an incidence angle of 8° and the reflectance measured diffusely, CIELab colour space with illuminant D65 and standard 10-degree observer.

## 4. Conclusions 

In this work, micro-Energy Dispersive X-ray fluorescence (µ-EDXRF) and Raman spectroscopy have been successfully used at first in the characterization of the pictorial layers of Ondarroa’s rotating walkway. Considering the lack of knowledge of the paints used for metal structures at the beginning of the twentieth century, a scientific diagnostic study was necessary. The samples collected from the two green pieces have made it possible to identify the original polychrome of the Ondarroa footbridge, *Plaiako Zubixe*, as well as the following applications of different pictorial layers. 

The original colour of the current green parts of the bridge was not green. It was a greyish blue colour that was obtained by mixing ultramarine blue, in very fine grains (~30/90 µm), with larger grains (~300 µm) of carbon black and lead white, dispersed in a greater amount of sulfate of barium, used as opacifier. The discovery of the original blue-grey layer is fully compatible with the black and white images of the catwalk, taken at the time of its inauguration.

However, despite the fact that the diagnostic analyses had shown that the current green colour did not correspond to the original, the inhabitants decided to restore the same colour that the bridge had in recent years. This work is a clear example of how the restoration of a work of art does not only concern the object itself. When it is part of a community or it is considered as a symbol of a city, the rules that are generally applied to the restoration do not apply when the opinion of an entire community prevails.

The bad oxidation state of the bridge has been the reason for its closure to the pedestrian and boat passage. Undoubtedly, the port marine environment in which it was constructed accelerated the oxidation state of the iron structure. This assumption was confirmed by the presence of iron degradation compounds, mainly akaganeite, which causes a cyclic alteration phenomenon that often ends with the total consumption of the iron core. In the same way, the identification of hematite and carbon demonstrated the SO_2_ impact from ships, road traffic, and industrial activity. Thus, the different iron oxides identified as corrosion products of the metal structure as well as the organic carotenoid (astaxanthin) acting as a bioindicator of the bad air quality, to which the bridge has been exposed since its construction, denotes the influence of both natural and anthropogenic factors in the state of conservation of the rotating bridge.

## Figures and Tables

**Figure 1 molecules-27-03609-f001:**
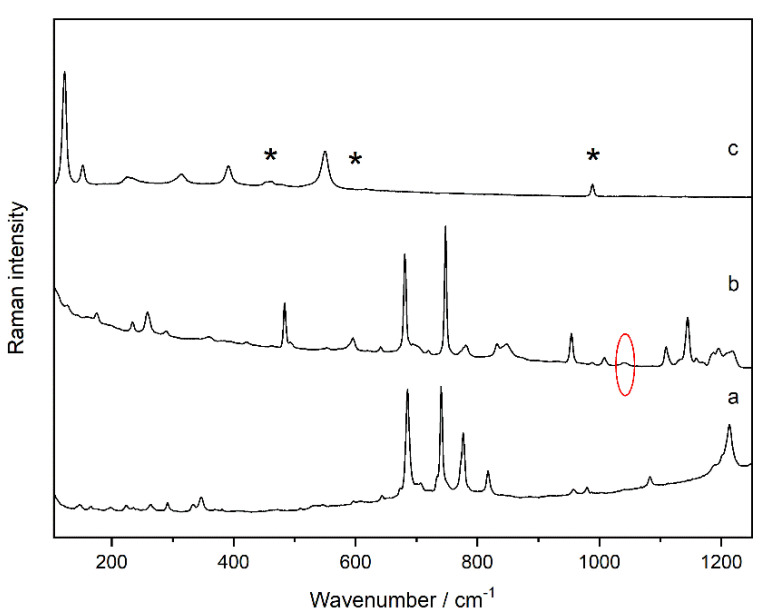
Raman spectra of the compounds identified in the subsample (SUBS-1a) in cross-section: phthalocyanine green (**a**), phthalocyanine blue with traces of azo pigment PY100 (in the red circle) (**b**), minium plus barium sulfate (*) (**c**).

**Figure 2 molecules-27-03609-f002:**
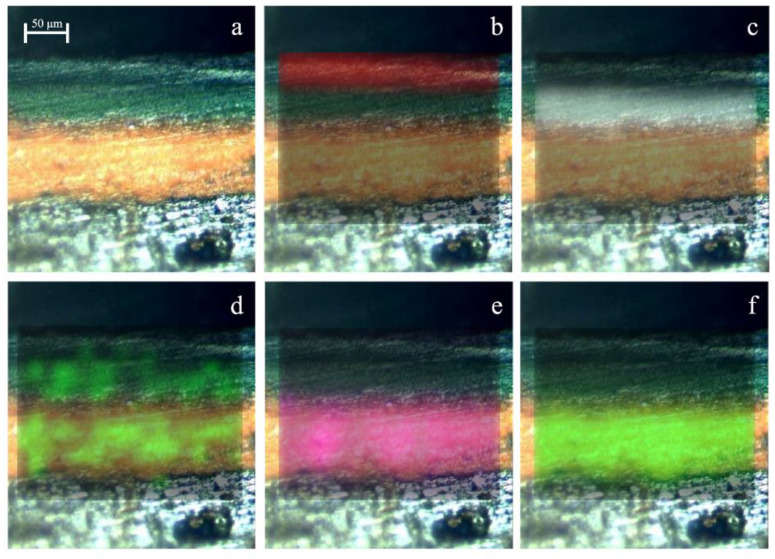
Raman image of a cross-sectional sample (sample SUBS-1a) (50×) collected from the railing piece (**a**) and its molecular composition: phthalocyanine green (**b**), phthalocyanine blue (**c**), yellow azo pigment (**d**), barium sulfate (**e**) and minium (**f**).

**Figure 3 molecules-27-03609-f003:**
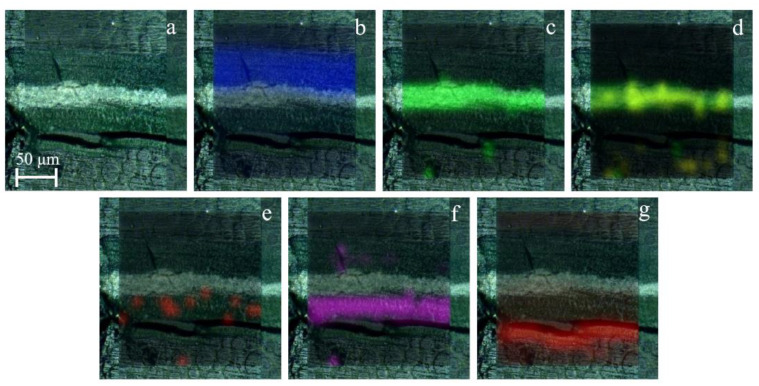
Optical image (**a**) and Raman spectroscopy image of the sample in section showing its molecular composition: phthalocyanine green (**b**), rutile (**c**) and calcium carbonate (**d**), barium sulfate (**e**), Prussian blue (**f**), and phthalocyanine blue (**g**). The black line through the sample is a crack in the sample.

**Figure 4 molecules-27-03609-f004:**
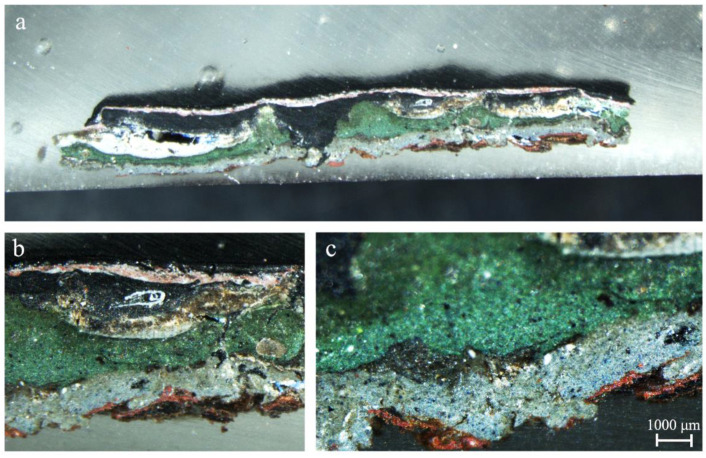
Cross-sections at diferent magnification (**a**–**c**) of the painting sample SUBS-3b from a black area where the original paint layer is visible.

**Figure 5 molecules-27-03609-f005:**
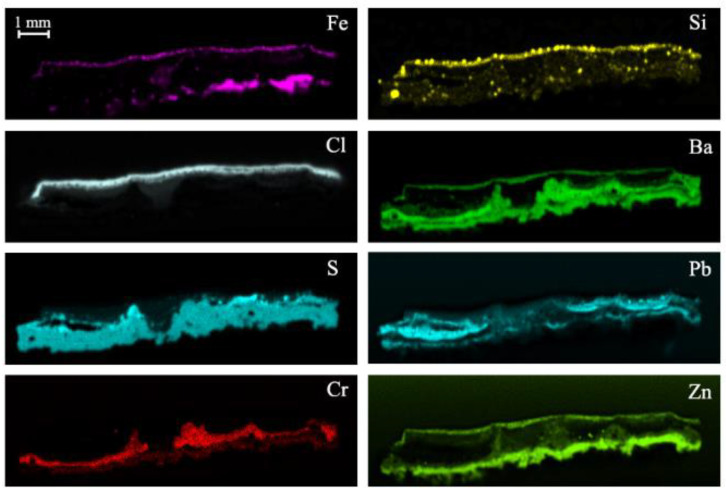
Micro-energy-dispersive X-ray fluorescence spectroscopy (µ-EDXRF) maps of the sample SUBS-3b from a black area.

**Figure 6 molecules-27-03609-f006:**
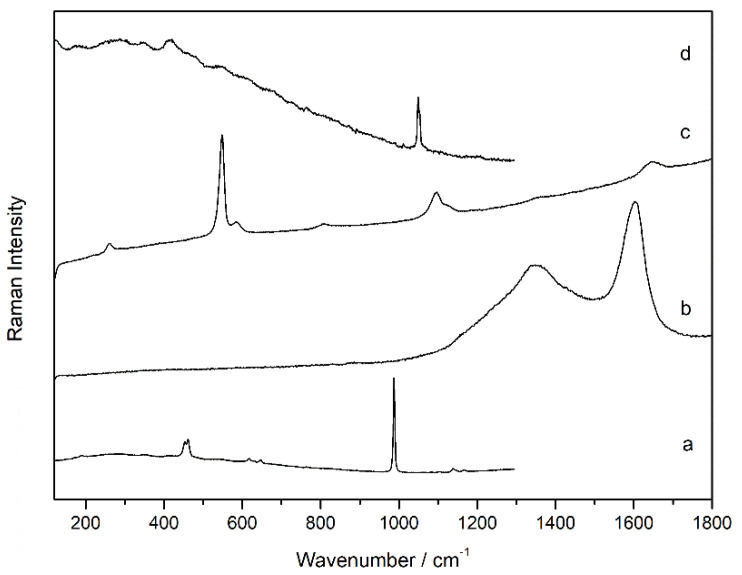
Raman spectra recorded in the original grey layer (sample SUBS-3b): barium sulfate (**a**), carbon (**b**), ultramarine blue (**c**), and lead white (**d**).

**Figure 7 molecules-27-03609-f007:**
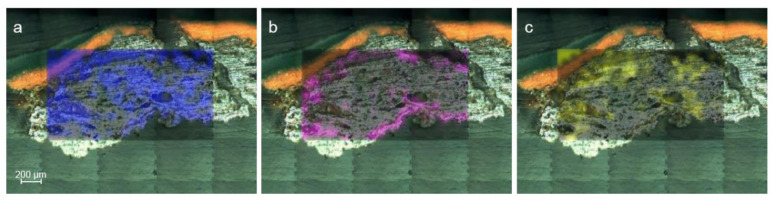
Raman images show the distribution of goethite (**a**), lepidocrocite (**b**), and magnetite (**c**) in the sample SUBS-2a.

**Figure 8 molecules-27-03609-f008:**
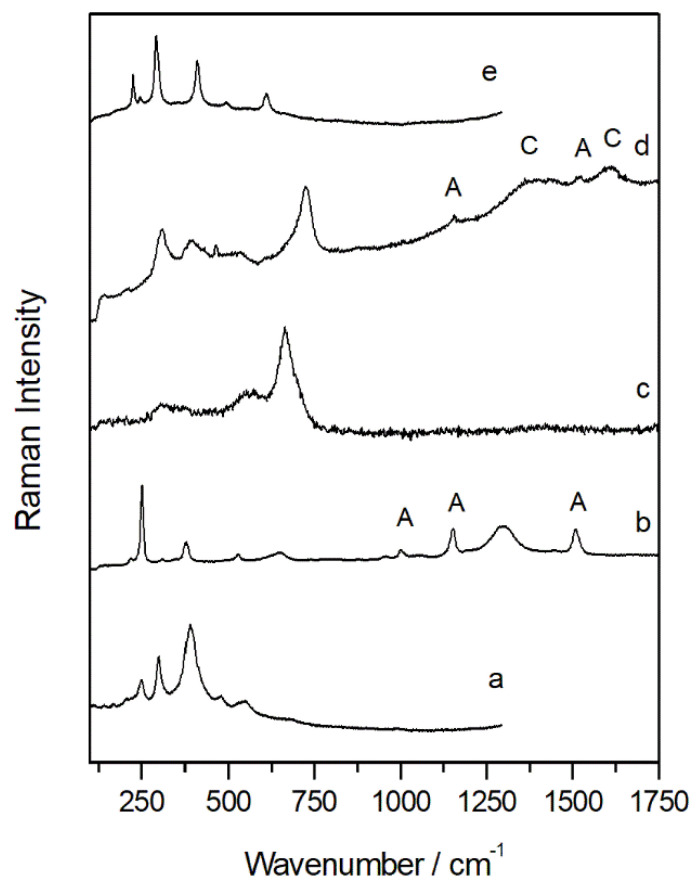
Raman spectra of goethite (**a**), lepidocrocite plus astaxanthin (A) (**b**), magnetite (**c**), akaganeite with traces of astaxanthin (A) and carbon (C) (**d**), and hematite (**e**) from the sample SUBS-4b.

**Table 1 molecules-27-03609-t001:** EDXRF elemental data (wt.%) of sub sample SUBS-3b.

Sample ID	Mg	Al	Si	S	Cl	K	Ca	Fe	Zn	Sr	Ba	Pb	Cr
SUBS-3b	1.63	0.78	3.4	7.2	16.4	0.3	10.8	12.8	3.44	0.98	28.2	13.9	0.2

**Table 2 molecules-27-03609-t002:** Lab values of the colours measured in the samples in Appendix A.

Colour	L *	a *	b *
Green (Appendix A)	34.96	−11.22	1.68
Greyish blue (Appendix A)	44.47	−0.14	2.07

***** The CIELAB color space is referred with asterisks to prevent confusion with Hunter Lab.

## Data Availability

The data supporting the findings of this study are available within the article.

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
