# Peer review of "Analytical Techniques Applied to the Study of Industrial Archaeology Heritage: The Case of Plaiko Zubixe Footbridge"

_molecules, 2022, doi:10.3390/molecules27113609_

Round 1

Reviewer 1 Report

I read the paper titled "Analytical techniques applied to the study of industrial archaeology heritage: the case of Plaiko Zubixe footbridge" I found the case study interesting although the analysis of painting is well known, the results provide an important diagnostic approach to make study o "new" field of cultural heritage: the industrial archaeology.
The paper is well written, the goal is focused and the manuscript is potentially interesting for the readers of Molecules.
As a reviewer, I still have some comments and suggestions.

1. authors affirm that "The original color of the current green parts of the bridge was not green. It was a greyish blue color that was obtained by mixing ultramarine blue, in very fine grains, with larger grains of carbon black and lead white" but how do you exclude that carbon black particles can come from the environmental pollution, accumulated between the first and the second painting?
2. the quality of Raman figures should be improved to make them easier readable, for this reason, iI suggest indexing the peaks already cited in the text for each spectrum.
3. please double check the references style ie. line 535

Author Response

Authors affirm that "The original color of the current green parts of the bridge was not green. It was a greyish blue color that was obtained by mixing ultramarine blue, in very fine grains, with larger grains of carbon black and lead white" but how do you exclude that carbon black particles can come from the environmental pollution, accumulated between the first and the second painting?

  • We think that carbon black is used as a pigment, and not that it is present as an external contamination as the black grains are mixed with the other pigments (see Figure a). Indeed, it is not form a layer above the paint layer as if it was a superficial deposit.

The quality of Raman figures should be improved to make them easier readable, for this reason, iI suggest indexing the peaks already cited in the text for each spectrum.

  • We have decided, for greater clarity in the reading of the Raman spectrums, to indicate with the initial of the name of the compound when they occur in a mixture. Otherwise, when dealing with a single compound we preferred not to insert anything because the bands are listed in the text. The insertion of the band numbers would result in a flattening of the spectra, for reasons of space in the graphs.

Please double check the references style ie. line 535

  • The reference has been corrected.

Reviewer 2 Report

The paper "Analytical techniques applied to the study of industrial archaeology heritage: the case of Plaiko Zubixe footbridge" by Ilaria Costantini, Kepa Castro, Juan Manuel Madariaga and Gorka Arana is an interesting and original study in the field of application of modern analytical research methods in the study of archaeological heritage objects. However, there are questions and comments on the text of the article that need to be corrected before publication.

  • Line 61: from the chemistry point of view, it sounds a little weird, it is clear what the authors had in mind, but it is one thing to talk about; chlorides dissolved in water that is present in the marine atmosphere, and another thing to talk about the presence of chlorides directly in the atmosphere;
  •  Line 66: authors talk about the formation of acids from acid gases. I would like to see literary sources where this information is taken from. In fact, the formation of these acids is a rather non-trivial process, in addition, carbonic acid is quite unstable and very quickly decomposes into carbon dioxide and water. Therefore, I would like to know exactly how the listed acids can be formed under such conditions;
  • Line 97: what is meant by colorimetric analysis - the generally accepted name for the method for determining the concentration of substances in solutions, or something else?
  • Table 1: total elemental composition over 100% - please check
  • Table 2: In the description of the table, it is desirable to add what kind of samples these are, so that you do not have to search in the text;
  • Line 448: "very fine grains" and "larger grains" - what exact values of grain size do the authors have in mind;

General remarks:

  • The numbering of the figures is rather strange, it is better to number sequentially 1-15;
  • The sequence of presentation of the material is unusual and does not contribute to its normal understanding. It is better to swap chapters 2 (Results and Discussion) and 3 (Materials and Methods), as is usually done
  • The standard method for identifying mineral dyes is X-ray phase analysis based on powder diffraction. In this article, conclusions on the phase and mineral composition are made on the basis of data on the elemental composition, which is not entirely correct. Why weren't powder diffraction experiments done?

I hope that the article will be published after making the neccesary corrections.

Author Response

Line 61: from the chemistry point of view, it sounds a little weird, it is clear what the authors had in mind, but it is one thing to talk about; chlorides dissolved in water that is present in the marine atmosphere, and another thing to talk about the presence of chlorides directly in the atmosphere;

  • This sentence has been corrected and the literary sources have been added in the text.

Line 66: authors talk about the formation of acids from acid gases. I would like to see literary sources where this information is taken from. In fact, the formation of these acids is a rather non-trivial process, in addition, carbonic acid is quite unstable and very quickly decomposes into carbon dioxide and water. Therefore, I would like to know exactly how the listed acids can be formed under such conditions.

  • The literary sources where this information is taken from have been added in the text In that paragraph.

Line 97: what is meant by colorimetric analysis - the generally accepted name for the method for determining the concentration of substances in solutions, or something else?

  • In this context, colorimetric analysis means the use of a tristimulus colorimeter to measure the color of a surface in the CIELAB color space, which is the reference standard for this kind of measurement and was defined by the International Commission on Illumination

Table 1: total elemental composition over 100% - please check

  • The values in the table have been corrected.

Table 2: In the description of the table, it is desirable to add what kind of samples these are, so that you do not have to search in the text.

  • The colorimetric analyzes were carried out on the larger samples delivered to the laboratory and not on the micro samples since the spot of the instrument measures 4 mm. Now, this detail has been added in the text.

Line 448: "very fine grains" and "larger grains" - what exact values of grain size do the authors have in mind;

  • The grain size has been added in brackets.

The numbering of the figures is rather strange, it is better to number sequentially 1-15;

  • The images that appear as an “S” prefix are those that will be included as supporting information and now they do not appear in the manuscript. The figures have been renamed in the text and those that belong to supporting information now do not appear in the text but are presented separately.

The sequence of presentation of the material is unusual and does not contribute to its normal understanding. It is better to swap chapters 2 (Results and Discussion) and 3 (Materials and Methods), as is usually done

  • Molecules Journal organizes scientific articles in this way. It is not our decision.

The standard method for identifying mineral dyes is X-ray phase analysis based on powder diffraction. In this article, conclusions on the phase and mineral composition are made on the basis of data on the elemental composition, which is not entirely correct. Why weren't powder diffraction experiments done?

  • X-ray diffraction was not used because all mineral dyes have been successfully identified thanks to the use of Raman spectroscopy and we thought that it was not necessary to use another technique. In addition, X ray diffraction is useful for crystalline compounds and in the samples under study, organic compounds mainly compose the paint layer.

Reviewer 3 Report

Manuscript Number: molecules-1739812

The manuscript “Analytical techniques applied to the study of industrial archaeology
heritage: the case of Plaiko Zubixe footbridge” presents study of 6 micro samples taken from Plaiko Zubixe footbridge to identify original color of the bridge as well as the impact of marine aerosol and harbor environment using mEDXRF and Raman spectroscopy and colorimetric analysis.

The topic is interesting; it is suitable for this journal and obtained results are valuable.

I would recommend this manuscript to be published in Molecules after following minor revisions:

Section 2:

  • Authors state: “All the samples collected in different areas in the largest piece, received in the laboratory…” Please specify the number of investigated samples.
  • Lines 114-124, Lines 154-159, Line 188, Lines 247-248, Lines 28-260, 265-269, 308-309, 311, 314-315: either cite appropriate references or show Raman spectra of pure compounds used for identification of different pigments.
  • Lines 118-119: please specify how “many Raman spectra” and collected from which samples?
  • Figure 1 caption: add explanation about red circle - what does it represent?
  • Figure 2 caption: what is show in figure 2a -add in the caption
  • Lines 148-149: reference should be cited which supports this claim “The use of red lead, currently banned due to its toxicity, was probably employed as antioxidant paint, following the rules in the second half of XX century.”
  • Line 150: I believe the usage of “the smallest sample” is confusing as there are only two metal pieces
  • Lines 242-243: please precisely specify samples. Also, it would be interesting to compare EDXRF results obtained from both investigated metallic pieces, i.e. from appropriate subsamples
  • Figure S7, Figure 6 and Figure 8 captions: specify which samples were investigated
  • Line 273: Sentence “Any chlorine compounds were found by Raman spectroscopy in this sample.: is not clear
  • Phrase “the oldest paint layer” used several time in the manuscript is not precise enough and should be changed.
  • Lines 304-306: sentence “The state of conservation of the pieces delivered to the laboratory was different from each other, therefore, an oxidized chip from the base of the railings (Figure S1a) and a sample with paint layer from the biggest one (Figure S1b) were selected.” is unclear. General comment: descriptions and marking of samples have to be improved.

Section 3.1.: Some comments on how representative are the two metal pieces of the condition of the entire bridge. Also it would be important to mark positions were samples were taken from in Figure S1. Based on what criteria only six samples were investigated? What was the size of the samples? How they were collected? Why five samples were investigated as cross sections and not six?

Specify which samples were investigated by which technique.

It would be good to improve English language throughout the manuscript.

Typos:

  1. Page 2, line 73, citation of reference [10]

Author Response

Authors state: “All the samples collected in different areas in the largest piece, received in the laboratory…” Please specify the number of investigated samples.

  • The number of the samples have been added

Lines 114-124, Lines 154-159, Line 188, Lines 247-248, Lines 28-260, 265-269, 308-309, 311, 314-315: either cite appropriate references or show Raman spectra of pure compounds used for identification of different pigments.

  • The bibliographic references used for the identification of the pigments have been included in the text.

Lines 118-119: please specify how “many Raman spectra” and collected from which samples?

  • This information have been added in the Materials and Methods section.

Figure 1 caption: add explanation about red circle - what does it represent?

  • In the caption, it has been added what the red circle indicates in figure 1.

Figure 2 caption: what is shown in figure 2a -add in the caption

  • In the caption, it has been added that figure 2a corresponds to the Raman image of the sample.

Lines 148-149: reference should be cited which supports this claim “The use of red lead, currently banned due to its toxicity, was probably employed as antioxidant paint, following the rules in the second half of XX century.”

  • The references have been added in this sentence.

Line 150: I believe the usage of “the smallest sample” is confusing as there are only two metal pieces

  • The sentence has been corrected in this way: Unlike the previous case, the subsamples that were collected from the smallest piece that permit the movement of the bridge (Figure S7b) showed a different stratigraphic composition from a first observation with the stereoscopic microscope

Lines 242-243: please precisely specify samples. Also, it would be interesting to compare EDXRF results obtained from both investigated metallic pieces, i.e. from appropriate subsamples

  • The name of the sample has been added to the sentence. We decided to carry out the micro XRF analyses only in the sample that had a more complex composition and structure. The samples taken from the metal piece of the railing have the same stratigraphic composition.

Figure S7, Figure 6 and Figure 8 captions: specify which samples were investigated

  • The name of the samples have been added in the captions

Line 273: Sentence “Any chlorine compounds were found by Raman spectroscopy in this sample. is not clear

  • The sentence has been corrected in this way: ”No chlorine compounds were found by Raman spectroscopy in this sample”

Phrase “the oldest paint layer” used several time in the manuscript is not precise enough and should be changed.

  • This description has been changed for the “original paint layer” or “the first layer applied”.

Lines 304-306: sentence “The state of conservation of the pieces delivered to the laboratory was different from each other, therefore, an oxidized chip from the base of the railings (Figure S1a) and a sample with paint layer from the biggest one (Figure S1b) were selected.” is unclear. General comment: descriptions and marking of samples have to be improved.

  • The mistake has been correct in this sentence.

Section 3.1.: Some comments on how representative are the two metal pieces of the condition of the entire bridge. Also it would be important to mark positions were samples were taken from in Figure S1. Based on what criteria only six samples were investigated? What was the size of the samples? How they were collected?

  • The samples have been taken using a scalpel at some points where different paint layers appeared. Now the sampling points are shown in Figure S7. The piece that at naked eye presented a greater chromatic heterogeneity was the sample that allows the rotation of the bridge. For this reason in this piece, 4 samples have been taken, especially in a slightly exposed part where traces of the original colour could remain. All samples have been treated as cross section but, the two samples of the railing piece were identical from a compositional point of view and therefore only the results of one sample were shown. We think that the samples were representative of exhaustive analyses since they are collected in different painted areas.

Why five samples were investigated as cross sections and not six?

  • This is a mistake. It has been correct in the text.

Specify which samples were investigated by which technique.

  • In table S1 in supporting information, the diagnostic techniques have been added.

It would be good to improve English language throughout the manuscript.

  • The English language has been revised

Page 2, line 73, citation of reference [10]

  • The mistake has been corrected